# Sub-picosecond collapse of molecular polaritons to pure molecular transition in plasmonic photoswitch-nanoantennas

Joel Kuttruff [1,11], Marco Romanelli [2,11], Esteban Pedrueza-Villalmanzo [3,4,11], Jonas Allerbeck [1,5], Jacopo Fregoni [6], Valeria Saavedra-Becerril[4], Joakim Andréasson [4], Daniele Brida [7], Alexandre Dmitriev [3] ✉, Stefano Corni [2,8] ✉ & Nicolò Maccaferri [7,9,10] ✉

Molecular polaritons are hybrid light-matter states that emerge when a molecular transition strongly interacts with photons in a resonator. At optical frequencies, this interaction unlocks a way to explore and control new chemical phenomena at the nanoscale. Achieving such control at ultrafast timescales, however, is an outstanding challenge, as it requires a deep understanding of the dynamics of the collectively coupled molecular excitation and the light modes. Here, we investigate the dynamics of collective polariton states, realized by coupling molecular photoswitches to optically anisotropic plasmonic nanoantennas. Pump-probe experiments reveal an ultrafast collapse of polaritons to pure molecular transition triggered by femtosecond-pulse excitation at room temperature. Through a synergistic combination of experiments and quantum mechanical modelling, we show that the response of the system is governed by intramolecular dynamics, occurring one order of magnitude faster with respect to the uncoupled excited molecule relaxation to the ground state.

Hybrid light-matter polaritonic states arise as a consequence of a coherent energy exchange between the confined electromagnetic field in resonators and the radiating transitions in molecules or, more in general, quantum emitters. The associated strong modification of the energy levels offers tantalizing opportunities of tuning various fundamental properties of matter, such as molecular chemical reactivity or electrical conductivity. As such, the so-called strong coupling regime holds a key potential in a broad range of fields, such as all-optical logic[1,2], lasing[3], superfluidity[4], chemistry[5], and quantum computing[6,7]. The fundamental requirement for strong coupling lies in boosting the light-emitter interaction to such an extent that the coherent energy exchange between light and emitters becomes greater than the individual decay rates. Such boost can be achieved by either resorting to a large number of emitters or by confining the electromagnetic field to sub-wavelength volumes. For the former, the most common resonators are photonic cavities, where the

[1]Department of Physics, University of Konstanz, Universitätsstraße 10, 78464 Konstanz, Germany. [2]Department of Chemical Sciences, University of Padova, via Marzolo 1, 35131 Padova, Italy. [3]Department of Physics, University of Gothenburg, Origovägen 6B, 412 96 Gothenburg, Sweden. [4]Department of Chemistry and Chemical Engineering, Chalmers University of Technology, Kemigården 4, 412 96 Göteborg, Sweden. [5]nanotech@surfaces Laboratory, Empa, Swiss Federal Laboratories for Materials Science and Technology, Überlandstrasse 129, 8600 Dübendorf, Switzerland. [6]Department of Physics, Universidad Autónoma de Madrid, Ciudad Universitaria de Cantoblanco, 28049 Madrid, Spain. [7]Department of Physics and Materials Science, University of Luxembourg, 162a avenue de la Faïencerie, L-1511 Luxembourg, Luxembourg. [8]CNR Institute of Nanoscience, via Campi 213/A, 41125 Modena, Italy. [9]Department of Physics, Umeå University, Linnaeus väg 24, 901 87 Umeå, Sweden. [10]Umeå Centre for Microbial Research, Umeå University, 901 87 Umeå, Sweden. [11]These authors contributed equally: Joel Kuttruff, Marco Romanelli, Esteban Pedrueza-Villalmanzo. ✉e-mail: alexd@physics.gu.se; stefano.corni@unipd.it; nicolo.maccaferri@umu.se

electromagnetic field is confined by metallic mirrors (e.g, Fabry-Pérot cavities[8] or multilayer heterostructures[9]), and polaritons emerge as collective excitations between the light modes and the ensemble of emitters (-10^6–10^10 molecules). For the latter, sub-wavelength confinement of light has been historically achieved by exploiting plasmonic architectures[10], and more recently also all-dielectric nanostructures[11–14]. In the context of plasmonics, near-field enhancement of the electromagnetic field is possible via localized surface plasmon resonances (LSPRs) with an effective mode volume of $1–100 \, nm^{3}$ [15,16], thus allowing the formation of polaritons even with a relatively limited number of emitters (from a single emitter to $10^3$). Due to the broad range of potential applications, the nature of polaritonic states formation and dynamics have been extensively researched over the last decade[17]. Coherent time-domain control of the reversible energy exchange between photons and matter, referred to as Rabi oscillations, has been demonstrated in J-aggregates and metal nanostructures[18], and for semiconductor quantum wells in a microresonator[19]. As well, various incoherent pathways to modify the polaritonic states on ultrafast timescales have been investigated, including charge transfer[20,21], saturation of semiconductor transitions[22], and ground-state bleaching in molecular systems[23–25].

The emerging branch of chemistry using strong coupling to modify chemical reactions is referred to as polaritonic chemistry[26]. There, polaritonic states have been applied to selectively suppress or enhance chemical reactions both in the ground and the excited states[27–30], opening up new chemical reaction pathways, including, among others, remote chemistry[31], singlet fissions[32], and selective isomerization[33–35]. Such manipulation of photochemistry makes use of the strong coupling with light to rearrange the electronic energy levels of molecules[36,37]. Achieving such control on ultrafast timescales promises many emerging applications combining ultrafast optics and light-driven chemistry[38]. This is exceptionally motivating in the context of molecular photoswitches that have already shown potential for inkless paper[39], stimuli-responsive materials[40], self-healing polymers[41], and all-optical switching[42], due to the ability to externally alter their molecular structure by light. One of the most famous photoswitches is spiropyran, which exist in a spiro (SP) and a merocyanine (MC) isomer. UV light triggers the SP→MC isomerization, whereas the reverse reaction is induced by visible light. These compounds display intriguing properties as ultrafast molecular photoswitches, due to the sub-ps kinetics associated with their molecular interconversion[43], and might be key elements to implement future all-optical molecular transistor technologies. The signature feature of the MC form is the emergence of a strong $\pi - \pi^*$ absorption resonance in the visible spectrum[44]. While the evidence that strong coupling with the $\pi - \pi^*$ transition modifies the SP-MC photoconversion rate is a milestone for polaritonic chemistry[36], the opportunities for this system at ultrafast timescales are still unexplored.

Here, we devise an archetypal platform capable of selectively accessing the weak and strong coupling regimes and follow the polariton dynamics after impulsive femtosecond-laser excitation. The platform consists in an array of two-mode anisotropic plasmon antennas and spiropyran photoswitches converting to MC form via continuous UV irradiation. To track the role of the coherent plasmon-molecules interaction on ultrafast timescales, we follow the time-evolution of polaritonic states with pump-probe experiments, where the dynamics of the strong coupling is referenced to the weak coupling in the exact same system. Quantum simulations, comprising a theoretical framework based on extending the original Tavis-Cummings Hamiltonian[45] allow us to interpret the experimental findings assuming changes in the fundamental properties of the coupled system. Our experimental and theoretical analysis reveals an ultrafast modification of the polaritonic state composition, identifying the main relaxation channel as the localization of the initial polaritonic coherent excitation on a molecular excitation. Intramolecular dynamics leads to sub-ps

changes of the polaritonic state manifold, one order of magnitude faster than expected from the pure transition time of excited molecules back to the ground state. Revealed sub-ps timescale control of the chemical energy landscape is crucial to advance polaritonic chemistry to the ultrafast regime.

## Results

We employ anisotropic aluminum nanoellipse antennas (see Methods and Supplementary Note 1 for fabrication details), displaying two orthogonal and spectrally separated LSPRs. A polystyrene nanofilm containing the spiropyran molecules (initially in the SP isomeric form) is spin-coated on top of the nanoantennas and subsequently irradiated with UV light to photo-isomerize the molecules to the MC configuration. The nanoantennas are designed such that the MC molecular absorption is resonant with the long-axis LSPR, whereas it is detuned from the short-axis LSPR, giving rise to strong and weak coupling regimes, respectively. The steady-state optical response of the hybrid system is shown in Fig. 1 for excitation along the long axis (sketched in Fig. 1a) and short axis (sketched in Fig. 1b) of the nanoantennas, respectively. While only the plasmon contribution is visible for the long axis case before UV irradiation (no MC isomers present, orange curve in Fig. 1c), lower (LP) and upper (UP) polaritonic states immediately emerge (black curve in Fig. 1c) upon UV-induced photoconversion of SP to MC, with the corresponding activation of the $\pi - \pi^*$ MC molecular transition. This is a characteristic signature of the system entering the strong coupling regime, or at least being at its onset[46–48]. Conversely, when excited along the short axis (Fig. 1d), the spectrum of the coupled plasmon-MC system (black curve) is governed by the superposition of LSPR (pink curve) and the molecular absorption. In this case, the plasmon with low extinction efficiency is out-of-resonance with the molecular transition, leading to two spectral peaks simply superimposing on each other. The absence of notable shifts of the absorption peaks with respect to the bare states in this case signals the weak-coupling regime. To distinguish between the two regimes, we analyze with a full quantum model the modification of the energies of molecular electronic states due to the interaction with the nanoantenna plasmon modes. First, in order to simulate the SP and MC linear absorption spectra (experimental spectra are shown in the inset in Fig. 1d, simulations can be found in Supplementary Note 2.1), the respective molecular ground state geometries are optimized (see Supplementary Fig. 5), and the vertical excitation energies to the first eight excited states of each isomer are calculated. The $S_1$ excited state of MC is then optimized at the same level of theory (in a solvent mimicking the polymer matrix), yielding the relaxed configuration of MC in the $S_1$ state minimum (see Supplementary Fig. 6). The electronic energy associated to this point is -0.4 eV lower than the vertical excitation energy at the ground state geometry, meaning that the molecule is not in its equilibrium configuration upon vertical excitation from $S_0$ to $S_1$. Interactions between different MC molecules and the nanoantennas is evaluated by means of an ad-hoc quantum model, which extends the original Tavis-Cummings (TC) Hamiltonian[49] by explicitly quantizing the modes of arbitrarily shaped nanoparticles[50]. The choice of this modeling strategy comes from the fact that the nanoantennas do not display any sophisticated geometrical features that may lead to "picocavity" formation[51] and the associated single-molecule strong coupling effects. Hence, investigating collective molecular phenomena while retaining the geometrical shape of the nanoantennas is required in this case. The calculations (Fig. 1e, f) corroborate the interpretation of the experimental spectra (more details on the theoretical model can be found in the Methods section and in the SI. In the latter, a quantitative discussion on the two coupling regimes is reported, see Supplementary Note 2.2).

After assessing that the system can enter the strong coupling regime by driving the long-axis LSPR, pump-probe measurements are performed on the hybrid system in both weak and strong coupling

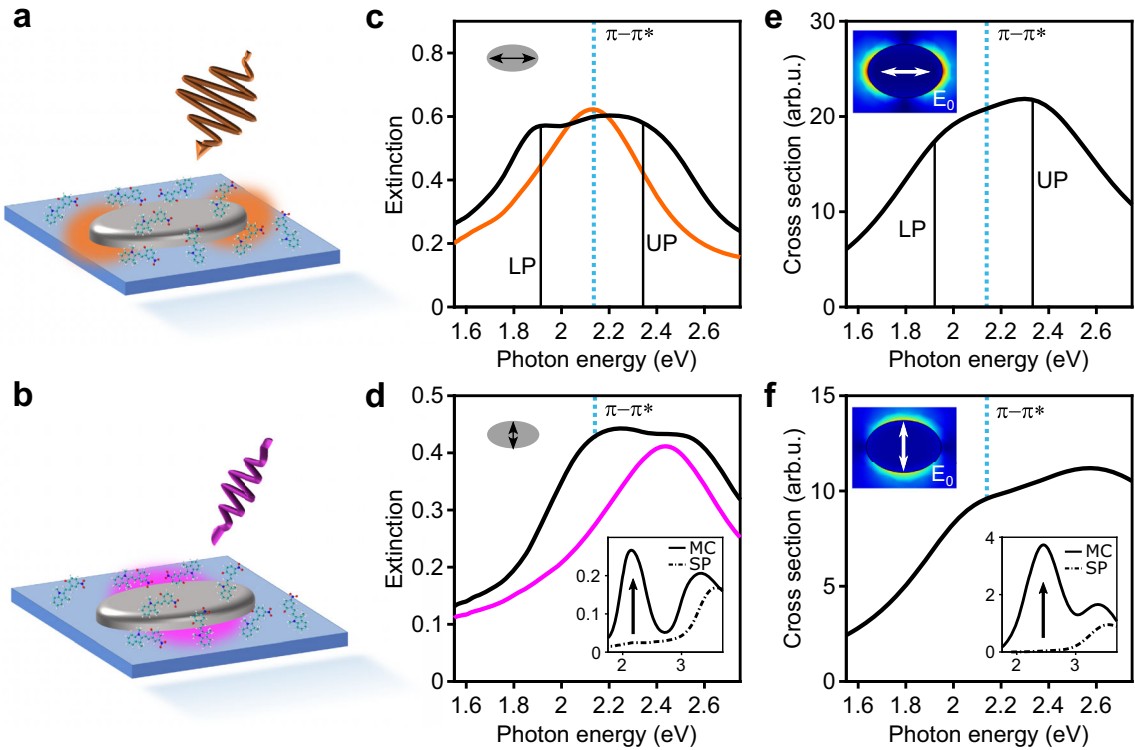

**Fig. 1 | Hybrid system consisting of photo-switchable molecules and aluminum nanoellipse. a, b** Sketch of the localized plasmon dipolar excitation along the long (**a**) and short (**b**) axes of the nanoantenna. The plasmon near fields are schematically indicated by hot spots (orange and pink) in the sketch. **c, d** Experimental extinction spectra for the long axis (**c**) and the short axis (**d**) before (colored) and after (black) UV-induced photoswitching of the spiro isomer (SP) to the merocyanine isomer (MC). The inset in **d** shows the absorption of a molecular film before (dash-dotted) and after (solid) UV irradiation, where the $\pi-\pi^*$ molecular absorption emerges at 2.15 eV (also indicated by the blue dotted lines). **e, f** Absorption cross sections obtained from our model Hamiltonian for long (**e**) and short (**f**) axis excitation of the hybrid system. Bottom inset in **f** shows the simulated molecular absorption cross section obtained after ground state optimizations of the molecular structure in the SP (dash-dotted) and MC (solid) configurations. Top insets show confined electric near fields at both plasmonic resonances calculated using the finite elements method.

regimes and referenced to a UV-exposed film (to trigger the SP→MC isomerization) of pristine spiropyran molecules, i.e., without the presence of the nanoantennas, as well as to the antenna array without UV switching the molecular film to the MC form.

The ultrafast response in all four cases (pristine molecules, antennas immersed in SP, molecules weakly coupled, and molecules strongly coupled to the nanoantennas LSPRs) is studied using a two-color pump-probe scheme based on a broadly tunable fs-laser spectroscopy platform operating at multi-kHz repetition rate[52]. Pump pulses with a temporal duration of 50 fs are tuned to photon energy of 2.3 eV, that is above the $\pi - \pi^*$ transition of the MC molecular form. Time delayed broadband probe pulses span a spectral range from 1.75 eV to 2.5 eV and are compressed to below 20 fs duration. The pump-induced change of probe transmission $\Delta T/T$ is monitored for various time delays $\Delta t$ between pump and probe pulses (see "Methods" section for more details). We first studied the transient transmission of the bare MC film (see Fig. 2). Figure 2a shows $\Delta T/T$ as a function of $\Delta t$ and the probe photon energy. For positive $\Delta t$, a broadband positive signal can be observed, indicating a transient bleaching of the molecular absorption, as previously observed for instance in ref. 53.

Such bleaching provides a direct measurement of the $S_1$ excited state population at each time, since less photons can be absorbed by the remaining molecules in the MC ground state. In Fig. 2b, we plot spectral cuts of the 2D map shown in Fig. 2a. A clear red-shifting of $\Delta T/T$ with increasing $\Delta t$ can be observed (see Supplementary Note 3.2. for more details). This can be explained by probe-stimulated photo-emission taking place during vibrational relaxation from the Franck-Condon point (where the molecule is vertically excited) toward the $S_1$

minimum, as sketched in Fig. 2c. The left panel in Fig. 2c shows the excitation of MC into the Franck-Condon configuration of the $S_1$ state. The molecular structure then relaxes to a lower energy configuration, subsequently leading to the observed redshift due to the stimulated emission back into the ground state (right panel). From the experiment, we find a maximum spectral shift of +0.25 eV, consistent with the value obtained theoretically (+0.4 eV; see also Supplementary Fig. 6).

In the pump-probe experiments on the antennas without UV-induced switching of the molecular film to the MC form, we see that the contribution of the antennas to the transient signal is at least one order of magnitude reduced compared to the response of the hybrid system (see discussion in Supplementary Note 3.3). This experiment is crucial to support that the effects observed in our experiments and reported hereafter really stem from the interaction of the plasmonic and molecular systems.

In Fig. 3, we show the ultrafast dynamics of the hybrid system upon light irradiation polarized either along the long or short axes of the nanoantennas, and using the same pump fluence as in the isolated molecules experiments discussed above. In general, sample geometry can lead to enhancement of certain spectral features in the ultrafast response of the specimen due to multiple interference, as observed for example in optical cavity strong coupling (see ref. 54.). In such cases, absolute absorption should be obtained via measuring both, transient reflection as well as transient transmission. In our simple geometry, however, interference is negligible in determining the ultrafast response and we can reliably use the transient change in transmission $\Delta T/T$ as a measure to follow the time-evolution of the polaritonic states. In particular, temporal and spectral dynamics are not affected by this choice.

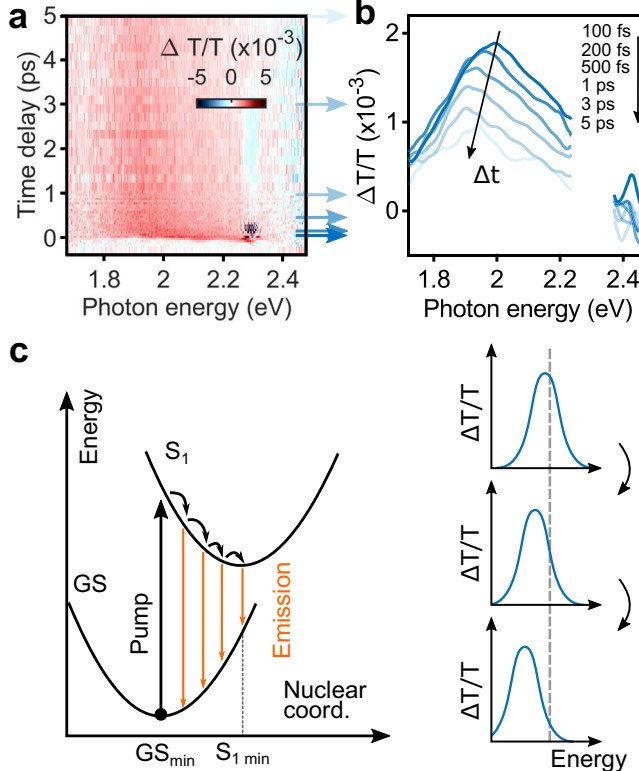

**Fig. 2 | Ultrafast dynamics of the MC isomer. a** Transient transmission signal as function of time delay between pump and probe pulses and probe photon energy. **b** Pump-probe spectra at increasing pump-probe delay (top to bottom) indicating a redshift of $\Delta T/T$. **c** Scheme of the wave packet relaxation dynamics on the $S_1$ excited state after the pump pulse. The red-shifting positive $\Delta T/T$ signal experimentally observed in panel **b** can be interpreted as stimulated emission from the $S_1$ surface while the system is relaxing towards the closest excited state minimum, $S_{1\ min}$ (left panel). Resulting $\Delta T/T$ spectra are schematically shown in the right panel.

The transient transmission for the long-axis pumping is shown in Fig. 3a as a function of $\Delta t$ and the probe photon energy. For positive time delays, we observe a characteristic positive-negative-positive (red-blue-red) spectral shape. Furthermore, our time and energy resolutions allow to track the spectral evolution of $\Delta T/T$, as shown in Fig. 3b, where spectral cuts of the pump-probe 2D map (Fig. 3a) are plotted for increasing values of $\Delta t$. As indicated by the blue dashed arrow in Fig. 3b, we observe a blueshift of around 20 meV of the negative $\Delta T/T$ peak within 1 ps after the optical excitation. The observed positive-negative-positive transient spectral lineshape was previously reported, for example, in molecules vibrationally coupled to a cavity[23], so called plexcitonic systems[55], or in an excitonic transition coupled to a plasmonic lattice[20]. It can be interpreted as a shift of the absorption peaks towards each other, and thus a reduction of the spectral separation of the UP and LP bands, seen as negative differential transmission around 2.1 eV. While this may be intuitively understood as a pump-induced decrease of the number of molecules effectively coupled to the mode[36,56,57] – thus resulting in a reduction of the Rabi splitting – explaining the spectral time evolution of $\Delta T/T$ as shown in Fig. 3b requires a more advanced analysis. After impulsive optical excitation, the upper polariton states of the hybrid system are populated and quickly dephase within few tens of femtoseconds, well below the time resolution of our experiment ($\approx 50$ fs).

We then assume that the molecule contributing most to the polariton (some molecules are more coupled to the modes than others, depending on their distance and orientation with respect to the nanoantennas) starts to relax on its $S_1$ excited state, eventually resulting in a system composed by $N$-1 molecules in their ground state

and one molecule vibrationally relaxing on its $S_1$ excited state. We remark that $N$ here should be interpreted as the number of molecules in strong coupling with the given plasmon mode, rather than all irradiated molecules (see also Supplementary Note 2.2). Being in their ground state, the $S_0 \rightarrow S_1$ electronic transition for the $N$-1 remaining molecules is still resonant with the long-axis plasmon; instead, the relaxation of the individual molecule on its excited state entails a redshift of the stimulated emission (we note that this is the typical composition that can be obtained for a dark state).

The polaritonic energy landscape is thus transiently modified by the intramolecular dynamics of the individual molecule, observable as a spectral time evolution of $\Delta T/T$ probed by the delayed optical pulse. To interpret the experimental results, we simulated the transient signal by taking the difference between the absorption spectrum of the 1-excitation-space polaritons (shown in Fig. 1), which is the transient spectral signature of ground state bleaching (GSB), and the signal coming from the localized red-shifting excited molecular state. The latter term requires computing both the stimulated emission (SE) to the ground state and the absorption toward the 2-excitation-space polaritonic manifold (excited state absorption, ESA)[58]. These quantities are then combined according to Eq.1 (see Numerical calculations section), yielding the transient data of Fig. 3c, f. (note that ESA and SE contributions do have opposite signs). Here, we perform the 2-excitation-space calculations (see Methods section for additional details) for increasing energy shift of the molecular transition, thus resembling the vibrational relaxation of the molecular structure. The energy shift values are based on the TDDFT results and consistent with experimental energy shifts of Fig. 2. The 2-excitation calculations have been repeated for different frequency shifts of the target molecule, and can reproduce the blueshift of the transient signal observed experimentally. By tracking the spectral position of the negative dip in the $\Delta T/T$ spectra (Fig. 3b, c), we obtain blueshifts of ~20 meV in the experiment and 25 meV in the simulation. The time dynamics of the relaxation process is not directly simulated, but is modeled by fitting the experimental data, as described in Fig. 3 and related discussion in the Methods section.

Moreover, we emphasize that the spectral blueshift of $\Delta T/T$ within the first picosecond is not an optically induced shift, but rather originates from a molecular excited state energy shift due to vibrational relaxation. Notably, simulating the transient signal via calculation of both the 1- and 2-excitation-space polaritonic manifold can reproduce well both the positive-negative-positive motive of the experimental transient spectra and the slight blueshift of the negative peak at increasing $\Delta t$, as we show in Fig. 3c. The observed spectral evolution of $\Delta T/T$ can therefore be reasonably assigned to a molecular vibrational relaxation that affects the composition of the polaritonic states manifold.

A simplified sketch describing the resulting rearrangement of electronic energy levels upon optical pumping is depicted in Fig. 4. The origin of the positive-negative-positive motive follows from what was already anticipated above: the arrival wavefunction following the probe absorption in our modeling approach is dominated by the product state of an excited molecule state times the 1-excitation polariton of the remaining $N$-1 molecules ($^2$UP/LP$_{N-1}$ in Fig. 4c). The corresponding UP-LP splitting is therefore reduced compared to the ground state absorption, thus leading to the characteristic differential positive-negative-positive $\Delta T/T$ transient spectral signature. This is schematically represented in Fig. 4d where the peak-to-peak separation of the blue curves is smaller than the corresponding one of Fig. 4b, the latter showing the linear absorption spectral shape. Besides, note that SE is providing a minor contribution to the signal (see Supplementary Fig. 13 for explicit comparison of GSB, ESA, and SE contributions).

Upon vibrational relaxation, the state before probe absorption (red energy level of Fig. 4c) is decreasing its energy faster than what the

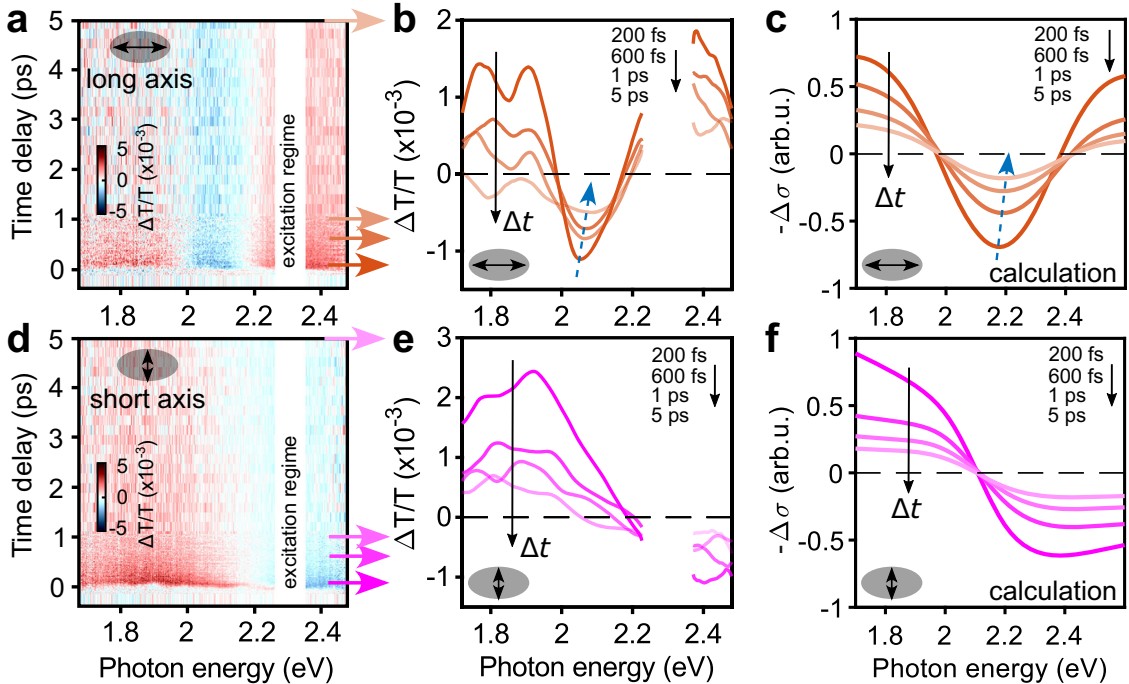

**Fig. 3 | Ultrafast dynamics of the hybrid system in the strong and weak coupling regimes. a** Transient transmission as function of time delay between pump and probe pulses and probe photon energy when excited along the long axis of the nanoantennas, where the longitudinal plasmon frequency is perfectly tuned to the excitation energy of the MC isomer. **b** Spectral cuts of the pump-probe map in **a** at 200 fs, 600 fs, 1 ps, and 5 ps. **c** Simulated transient response for the long-axis pumping case. We highlight that the plotted quantity -Δσ, which is the change in absorption cross section (with a minus sign) upon pumping, corresponds to transient transmissivity up to a constant multiplicative factor. Spectral dynamics are well reproduced by red-shifting the molecular transition of one MC (see sketch in Fig. 4c) according to vibrational relaxation of the molecule. The energy shift values

employed correspond to those obtained at 200 fs, 600 fs, 1 ps, and 5 ps after excitation (see numerical calculations section for more details on this energy shift-to-time delay connection). **d–f** Transient transmission, spectral cuts and simulated transient response for the excitation along the short axis of the nanoantennas, where the MC isomer are off to the plasmon resonance and one of them is gradually red-shifted by the same values as in **c**. Dashed line is the zero-crossing of the transient signal. In this case, the excited state relaxation does not influence the dynamics of the weakly coupled system, due to the large initial frequency mismatch with the plasmon resonance. The corresponding simulated 2D maps can be found in Supplementary Fig. 12.

arrival 2-excitation states do ($^2$UP$_{N-1}$ and $^2$LP$_{N-1}$ of Fig. 4c) leading to a slight blueshift of the transient signal (see the blue-shifting ESA contribution of Fig. 4d whose time evolution is schematically indicated by the blue arrow and corresponding dashed curve). This is reasonable, since the starting state is essentially a localized excited state of the molecule, while in the arrival state the excitation is partly delocalized on vibrational unrelaxed molecules, with higher energy. On a side note, we observe that the magnitude of the simulated $\Delta T/T$ signal increases over time, in contrast to what is being observed experimentally. Such discrepancy is expected and is explained by the lack of population decay in the simulations, which physically may be caused by either stimulated emission or non-radiative decay to the ground state. To phenomenologically account for this signal decay, we added an exponential decay on top of the simulated signal (details can be found in the Numerical calculations section), thus resulting in the simulated spectral cuts of Fig. 3c, f that indeed feature a decreasing magnitude over time.

In Fig. 3d, the transient transmission in the weak coupling regime (i.e., for the short-axis pumping system) is shown. In contrast to the strong coupling case, we now observe a differential (positive-negative) transient spectral lineshape, indicative of a shift in linear absorption of the system. Spectral cuts are shown in Fig. 3e. There are spectral changes of the differential signal over time, however, no clear spectral shift can be identified.

The same differential signal is again well captured by the theoretical simulations shown in Fig. 3f. In this case, due to the initial frequency mismatch between the short-axis nanoantenna resonance and the MC transition, the pump quickly drives the system towards a

regime where only $N-1$ remaining MCs are actually able to couple with the short-axis plasmon mode, already at early times, thus reducing the redshift of the molecular absorption caused by the weak coupling with the plasmon compared to ground state absorption, leading then to the positive-negative spectral feature observed. We remark that even in the weak coupling regime the signal that is being observed is not coming from mere isolated molecules, but it is rather originated by the modified molecular and plasmon responses because of the locally enhanced plasmonic field.

We now discuss the timescales of the observed ultrafast dynamics in more detail (see Fig. 5). The transient transmission of the bare MC film is shown in Fig. 5a, where the signal at the probe photon energy of the $\pi - \pi^*$ transition (2.1 eV) is shown by the red circles. The bi-exponential behavior follows from an ultrafast (on femtosecond timescale) spectral shift of $\Delta T/T$ and a subsequent picosecond decay. More in detail, the induced stimulated emission from the $S_1$ state quickly redshifts due to vibrational relaxation of the molecules from the Franck–Condon configuration towards the $S_1$ minimum, leading to a sub-ps (200 fs) change of $\Delta T/T$ at fixed probe photon energy. At lower probe photon energy of 1.85 eV (blue circles), corresponding to the excited state $S_1$ minimum, only a slower (on picosecond timescale) decay of $\Delta T/T$ is observable, related to the transition of the molecules back to the ground state, with a time constant of 12 ps. Interestingly, in the weak coupling case (Fig. 5b), a sub-ps decay appears at the relaxed transition energy of 1.85 eV. This can be inferred to an initial plasmonic component of the decay, which is due to the weak interaction of the molecular transition with the plasmon. After the localization of the wavepacket on the

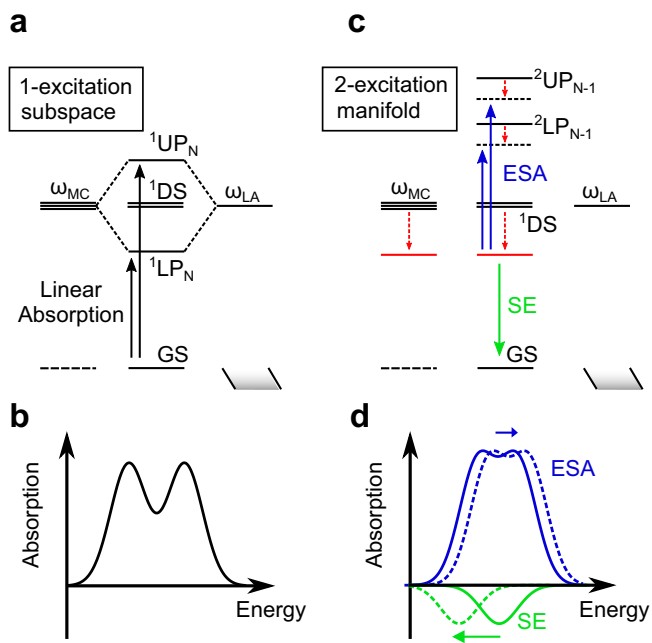

**Fig. 4 | Visual representation of electronic state reconfiguration after optical pumping in the strong coupling regime. a** Sketch of the electronic energy levels according to the theoretical model before optical pumping of a merocyanince (MC) sample strongly coupled to the long-axis plasmon. The 1 superscript in the state labels refers to polaritonic states belonging to the 1-excitation subspace obtained through diagonalization of the Hamiltonian in the 1-excitation basis. $^1$DS represent molecular dark states. **b** Schematic representation of the linear absorption corresponding to the energy levels shown in panel **a**. **c** Sketch of the electronic energy levels after optical pumping. The pump mostly populates $^1$UP at early times, which quickly dephases (≈ tens of fs), collapsing to a single localized MC excited state, which was originally contributing most to $^1$UP state (red energy level). Subsequently, that state undergoes vibrational relaxation (red dashed arrows), during which the probe can promote both stimulated emission (SE) to the ground state and excited state absorption (ESA) to the 2-excitation manifold. The two bright states of that manifold accessible from the localized state, here labeled as $^2$UP$_{N-1}$ and $^2$LP$_{N-1}$ and referred to in the text as arrival wavefunctions following probe absorption, can be mostly seen as product states composed of the localized excited state and the 1-excitation UP and LP states due to the other $N$–1 remaining molecules. Indeed, the energy of those states is less affected by the vibrational relaxation (the magnitude of the dashed arrows represents the relative energy evolution over time). **d** Schematic representation of the ESA and SE contributions (here shown separately for the sake of clarity) originating from the localized state, and their changes over time indicated as dashed curves. In order to obtain the transient signal $\Delta T/T$ (Fig. 3c), the sum of the ESA and SE spectra is subtracted from the computed linear absorption.

molecular states, the system decays in the same way as the bare molecules. By subtracting the response of the unperturbed molecules (see Supplementary Note 3.4. for more details), we can extract this ultrafast component induced by the weak coupling of the molecular transition to the plasmonic excitation (blue diamonds in Fig. 5b), yielding an exponential relaxation with time constant of 400 fs (dashed line in Fig. 5b). In the strong coupling case (Fig. 5c), the same analysis reveals the ultrafast response at the UP (upper panel in Fig. 5c) and LP (lower panel in Fig. 5c) hybrid states, yielding time constants of 350 fs and 360 fs, respectively. The acceleration of excited state decay both in the weak and strong coupling case compared to the bare molecules most likely occurs by plasmonic interaction via enhancement of non-radiative decay channels, as well as Purcell-like enhancement of radiative decay (see Supplementary Note 2.3 for a theoretical analysis). It should be noted however, that at this point we cannot quantitatively distinguish between non-radiative and radiative decay contributions in our experiments.

In their seminal work on spiropyran molecules coupled to optical cavities, Hutchison et al.[36] observed a quick decay of the upper polariton state to the lower polariton state and a longer-lived population of the lower polariton of several picoseconds. In contrast, we observe the same decay constants at the LP and UP spectral position. In agreement with the theoretical model (Fig. 4), this suggests that we merely observe the decay of the molecule excited state (thus affecting the overall ultrafast response), that is accelerated due to interaction with the metal nanoantennas. This apparent discrepancy between the two experiments can be explained by the poor quality factor (and thus higher damping) of plasmon modes as compared to cavity modes, resulting in shorter lifetimes of the coherent hybrid excitations, as for example shown in[22]. Thus, our combined analysis finds a dominant role of the accelerated intramolecular dynamics on the ultrafast response of the hybrid photoswitch-nanoantennas that is at least one order of magnitude faster than is expected from the transition of uncoupled excited MC molecules back to the ground state. Therefore, this is the right timescale to be probed to reveal strong-coupling effects on molecular photochemistry and photophysics.

## Discussion

We presented an in-depth study combining advanced quantum modeling and pump-probe spectroscopy of light-matter coupling in a prototypical photoswitchable molecular-plasmon system stably working at room temperature. The optical anisotropy of the plasmonic resonator allows simultaneous access to weak and strong coupling regimes, depending on the incoming light polarization with respect to the nanoantennas two main axes. In both weak and strong coupling regimes, experiments show the potential to affect the chemical energy landscape of the hybrid system on sub-picosecond timescales, observing a significantly faster dynamics of the excitation with respect to the purely molecular relaxation time. We demonstrate that this effect cannot be explained by a simple plasmon non-radiative decay, which occurs in the first few tens of femtoseconds, but rather originates from the complex intramolecular dynamics within the coupled plasmon-molecules system. A quantum model based on the extension of the Tavis-Cumming Hamiltonian, which can capture the ultrafast dynamics in both the weak and the strong coupling regimes, was developed. This theoretical framework allows also to closely map the ultrafast dynamics to changes of the electronic states in the hybrid system. Our synergistic approach combining ultrafast spectroscopy and advanced quantum modeling paves the way for deep understanding of the ultrafast dynamics in coupled plasmon-molecules systems in general. We believe that our results provide exciting foundation for the further exploration of the synthesis and characterization of strongly coupled photoswitch systems, towards a full control of ultrafast chemical processes at the nanoscale.

## Methods

### Fabrication and steady-state optical characterization

Aluminun nanoellipses are fabricated via hole-colloidal lithography[59]. Briefly, a sacrificial layer of PMMA is spin-coated on a quartz substrate, with a thickness around 250 nm. After depositing polystyrene (PS) beads of 100 nm diameter, a Cr mask is evaporated on top while the sample is tilted 45° with respect to the surface, allowing to cast a shadow on the PMMA surface with an elliptic shape. After tape-stripping the beads from the surface, $O_2$ plasma etching generates holes that allow the deposition of Al by e-beam evaporation. The final metasurface sample is obtained by lifting off the PMMA with acetone in an ultrasonic bath.

SP molecules were mixed with PS polymer in toluene and spin-coated on top of the Al metasurface and leave to evaporate at room temperature, forming a thin film covering the Al ellipses. More details regarding the fabrication are reported in the Supplementary Information.

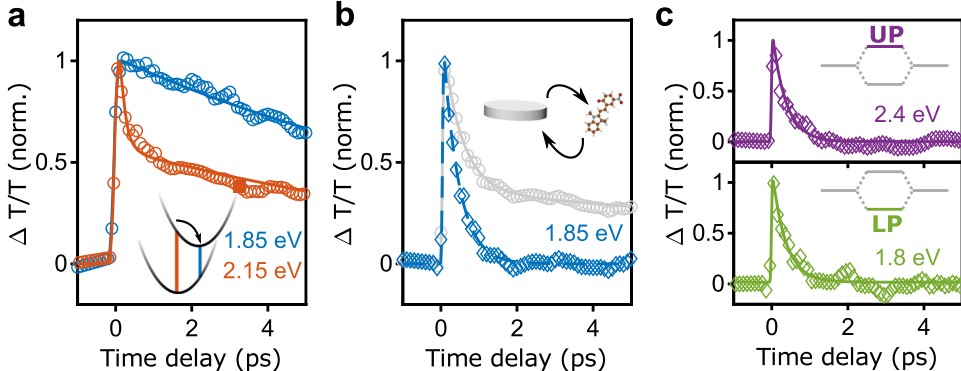

**Fig. 5 | Ultrafast dynamics of molecular and polaritonic states. a** Time-resolved cut of the 2D pump-probe data of the bare merocyanine reference film at the energy of the steady-state molecular transition, 2.15 eV (orange), and at the $S_1$ excited state minimum, 1.85 eV (blue). Ultrafast component corresponds to the vibrational relaxation of the molecule, while the slower picosecond component is related to the transition back to the ground state. **b** Cut for the hybrid system in the weak coupling case at the $S_1$ minimum. The gray curve is the raw data and blue curve is the remaining femtosecond decay, obtained by subtracting the slower component shown in **a**. **c** Ultrafast decay in the strong coupling case at the energy of the upper (2.4 eV) and lower (1.8 eV) polariton states, respectively. The slow component has been subtracted as in **b**.

Steady-state optical characterization shown in Fig. 1c, d was performed with white light polarized along the LA and SA of the Al nanoellipses, respectively. To convert the SP to its MC configuration, the sample was continuously illuminated with UV light (365 nm). The photoconversion was monitored by measuring the absorption at the $\pi - \pi^*$ transition (2.1 eV).

### Pump-probe spectroscopy

The experimental platform used in this study is based on a commercial Yb:KGW regenerative amplifier system working at a laser repetition rate of 50 kHz. Pump pulses are generated by a noncollinear optical parametric amplifier (NOPA) working in the visible spectral range, where a bandpass filter is used before the parametric amplification to restrict the spectrum to a central photon energy of 2.3 eV and spectral bandwidth of 50 meV. The pump-induced change of transmission is probed by a white-light supercontinuum generated from sapphire between 1.75 eV and 2.5 eV, which is temporally compressed using dielectric chirped mirrors. The pump pulse energy is set to 20 μJ/cm² and the probe pulse energy is adjusted to ensure at least a 1:10 energy ratio compared to the pump. Using a spherical mirror with 300 mm focal length, spot sizes of 120 μm and 150 μm are set for probe and pump pulses, respectively. Pump and probe pulses have parallel polarization and interact with the sample at a small noncollinearity angle, allowing to spatially block the pump pulse after sample interaction. Remaining scattering from the pump is then removed from the data in post-processing. A fast spectrometer camera spectrally resolves the probe pulse after sample interaction. For all measurements, the dye molecules are prepared in their MC configuration, as described above. Photoisomerization of some MC molecules during the pump-probe measurements is taken into account (see Supplementary Note 3.1 for more details).

### Numerical calculations

Quantum mechanical calculations of isolated bare SP and MC molecules were performed by means of Gaussian 16[60], the solvent was considered as an implicit medium through the default Gaussian implementation of the IEF-PCM formalism[61]. The choice of using ethylbenzene as implicit solvent to reproduce the polystyrene environment was done because of its close structural resemblance to the styrene molecule (they also present almost identical dielectric constants) and its prompt availability in the Gaussian package. The aluminum nanoellipse was created through the Gmsh code[62], (see Supplementary Fig. 7) and the coupling numerical values were computed through our homemade code TDPlas[50]. More precisely, one single nanoparticle (considering the elliptical shape, see Supplementary Fig. 7) is surrounded by many MC molecules following an elliptical grid pattern (shown in Supplementary Fig. 8), where each molecule is described as a point dipole oriented perpendicularly to the metal surface. Unlike the standard TC Hamiltonian, here we do not assume a constant coupling. Instead, we evaluate the coupling of each molecule with each plasmon mode, depending on its position and orientation w.r.t. the electromagnetic field associated to the plasmonic modes. In addition, we consider simultaneously the two (LA and SA) plasmon modes to include the two individual field distributions (electric near-field calculated via finite elements method shown inset of Fig. 1e, f). The coupling values for each molecule are numerically evaluated considering a Drude-Lorentz quantized description of the metallic response[50] and are then used to set up a Hamiltonian (see Eqs. 7–8, Supplementary Note 2.2) through which the steady-state absorption spectrum can be simulated. The eigenvalues and eigenvectors of the resulting Hamiltonian, that is, the energies and state composition of the polaritonic states, are then used to obtain the steady-state (linear) spectra by making use of a linear response expression of the polarizability, evaluated as a sum over the polaritonic energy levels. The same Hamiltonian written in the 2-excitation space (see Supplementary Note 2.2) is then diagonalized to obtain information about the 2-excitation-space polaritons and simulate the transient signal observed in the pump-probe experiments shown in Fig. 3, according to:

$$-\Delta\sigma = GSB - (ESA + SE), \tag{1}$$

where the explicit theoretical expression for computing each term of Eq. 1 is described in the SI (Supplementary Note 2.2).

The simulated raw data for the LA-SA case of Fig. 3c, f featured constant deviations from the zero baseline, so a-posteriori constant shifts have been applied to each transient spectrum to obtain the data reported in Fig. 3c, f. More precisely, each simulated curve has been overall positively shifted to make its inflection point aligned with the zero baseline. Moreover, to quantitatively estimate the time scale of the vibrational relaxation dynamics, and thus allowing us to have a 1-to-1 mapping between molecular energy shifts (which are the actual input in the calculations) and time delays reported in Fig. 3c, f, we fit the experimental energy shift of bare relaxing MC molecules (see Fig. 2 and Supplementary Note 3.2) to obtain the time constant of the vibrational relaxation. We then express the energy shift as $\Delta\omega(t) = \Delta\omega_{max}(1 - e^{-t/\tau_2})$ with $\Delta\omega_{max} = 200$ meV and $\tau_2 = 200$ fs, as shown in Supplementary Fig. 19. We note that this procedure does not take into consideration that pump and probe pulses have a finite

temporal width and that even polaritons, which are formed at early times, also feature a small-yet-finite lifetime. Both these contributions, that we are neglecting in the fitting procedure, may slightly affect the connection between molecular energy shifts and time delays that we are making use of in Fig. 3c, f. We remark that this 1-to-1 mapping is used to facilitate the comparison with the experimental data, creating a bridge between the molecular energy shift and time delay.

In addition to that, as described in the main next, the lack of population decay in the simulations leads to an ever-increasing magnitude of the $\Delta T/T$ simulated signal, so to phenomenologically account for such missing feature we added an exponential decay on top of the simulated data like $e^{-t/\tau_1}$ for $t < 1$ ps with $\tau_1 = 400$ fs (based on the experimental signal). Moreover, since the experimental data of Fig. 3a, d presents a constant magnitude after $\approx 1$ ps that is roughly 1/5 of the intensity at early times, the same constant feature has been applied to the simulated data to recover a similar spectral trend at longer times (>1 ps).

Similar approaches that use fitted rates and phenomenological decays in theoretical models to better comprehend experimental signals have been widely used before[63–67].

As a final note, the linewidths (decay rates) associated with each polaritonic transition that are required for the simulated spectra are obtained by resorting to a non-Hermitian formulation of the Hamiltonian where energies of the uncoupled states bring an imaginary component which corresponds to either a molecular or plasmonic decay rate, corresponding to vibrational broadening and plasmon damping, respectively (see SI for the values considered)[68,69]. The diagonalization of such Hamiltonian directly returns the decay rates of the polaritons as imaginary component of their associated eigenvalues[70]. We highlight that the decay rates employed in the Hamiltonian are used to account for the broadening of the transitions (corresponding to a lifetime of $\approx$ few fs for the plasmon linewidth and $\approx 100$ fs for the molecular transition, see SI), whereas the phenomenological decay previously mentioned ($\tau_1 = 400$ fs) was used to account for population decay (i.e. signal intensity decrease) over time because of non-radiative decay. Explicit theory formulation can be found in the SI.

## Data availability
The authors declare that the data supporting the findings of this study are available within the paper, its supplementary information files, and from the corresponding authors upon request.

## Code availability
The code used for the simulations is available upon reasonable request from M.R. and S.C.

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

## Acknowledgements

N.M. and D.B. acknowledge support from the Luxembourg National Research Fund (Grant No. C19/MS/13624497 'ULTRON'). N.M., D.B., and S.C. acknowledge support from the European Union under the FETO-PEN-01-2018-2019-2020 call (Grant No. 964363 'ProID'). D.B. acknowledges support from the European Research Council through grant no. 819871 (UpTEMPO) and from ERDF Program (Grant No. 2017-03-022–-19 'Lux-Ultra-Fast'). J.K. acknowledges the German Research Foundation via SFB 1432. A.D. acknowledges the Swedish Research Council (Grant No. 2017-04828) and Swedish Research Council for Sustainable Development (Formas) (Project No. 2021-01390). N.M. acknowledges support from the Swedish Research Council (grant no. 2021-05784), Kempestiftelserna (grant no. JCK-3122) and the Wenner-Gren Foundation (grant no. UPD2022-0074). N.M. and S.C. acknowledge from the European Innovation Council (grant n. 101046920 'iSenseDNA').

## Author contributions

J.K. designed and performed the pump-probe experiments with the support of J.A. (Jonas Allerbeck). M.R., J.F., and S.C. developed the theory and performed the quantum simulations. E.P.-V. and A.D. designed and fabricated the plasmonic photoswitch-nanoantennas and performed steady-state optical, chemical, and photochemical characterization with the support of V.S.-B. and J.A. (Joakim Andréasson). D.B. contributed to the data analysis and discussion. N.M. conceived the idea and supervised the work. J.K., M.R., E.P.-V., A.D., S.C., and N.M. analyzed and discussed the results, and wrote the manuscript with input from all the authors.

## Funding

## Competing interests

The authors declare no competing interests.
