## [Peer Review File · Nature Communications]

Sub-picosecond collapse of molecular polaritons to pure molecular transition in plasmonic photoswitch-nanoantennasREVIEWER COMMENTS

Reviewer #1 (Remarks to the Author):

The article studies the ultrafast dynamics of spiropyran/merocyanine molecules coupled to aluminum ellipse nanoantennas through pump-probe spectroscopy. Depending on the input polarization with respect to the two main axes of the nanoantenna, the plasmonic resonance can match or not the molecular absorption peak, leading to strong or weak coupling regimes, respectively. Quantum theoretical modeling is used to describe the obtained results, in terms of the evolution of the polaritonic state composition of the hybrid system. The manuscript is in general well written, and I believe presents a very interesting and original contribution to the field. There are, however, some issues that should be addressed before publication, which I specify below.

Comment 1: The introduction refers only to plasmonics as a mechanism to confine the electromagnetic field beyond the diffraction limit to boost light-emitter interactions. I suggest that the authors also briefly mention relevant literature on dielectric nanoantennas in this area (see for example, Nano Lett. 16, 8, 5143–5151, 2016; Nano Lett. 17, 1219–1225, 2017).

Comment 2: An important experiment missing is the dynamic response of the bare plasmonic nanoantennas, as pump-probe results are shown only for the hybrid system or the bare molecular film. It is important to have separately characterized all individual elements composing the hybrid system. Can the authors comment on this?

Comment 3: Does the signal of the hybrid system degrade over time? (On a timescale of minutes, hours, days...), i.e., because of light induced damage or permanent modification of the molecule.

Comment 4: In reference to Figure 3c, the authors affirm that “theory can reproduce well both the positive-negative-positive motive of the experimental transient spectra and the slight blue shift of the negative peak at increasing Δt ”. However, I cannot see any blueshift in this figure. I suggest that the authors explicitly mention and compare in the main text the magnitude of the experimental and theoretical shifts.

Comment 5: Regarding Figure 3e, the authors state that “there is no significant spectral evolution of the signal over time”. However, it can be clearly seen that the spectral shape does significantly change over time. This sentence should be better expressed.

Comment 6: The authors explain the observed frequency shift in the hybrid system as a combination of excited state absorption and stimulated emission of the relaxing molecule. I suggest that they also mention the relative contributions of these two components, required to adequately describe their experimental findings.

Comment 7: Related to my previous comment, I further recommend that the authors revise the text discussing the observed energy shifts over time, as it can be complicated to follow at parts (especially in the analysis in relation to Figures 3 and 4).

Minor comments:

i) The authors refer several times in the text to Figs. 3c,f as Figs. 3c-f (note that this can be interpreted as all panels between c and f, and not just c and f).

ii) In reference to Figure 5, in the main text, it says "...blue circles in Fig. 5a...". However, there are no blue circles in Fig. 5a (only blue diamonds or grey circles).

iii) To facilitate understanding, it would be convenient that the authors add more information of the measurement conditions to the individual graphs. For example, so that one can understand that Figure 3 a,d corresponds to mutually orthogonal polarizations without referring to the caption. The same applies to Figure 5a, which shows plots at different probe energies, but this is not specified in the graph. This can be extended to other figures in the article.

Reviewer #2 (Remarks to the Author):

The authors study the ultrafast dynamics of films of spiropyran-merocyanine photochromes dispersed in a polymer film over elliptical aluminium plasmonic particles, for which the long axis plasmon mode is resonant with the merocyanine (MC) and for which the short axis plasmon is shifted to be less resonant.

They find significant changes in the transient spectroscopy and dynamics for the putative strongly coupled system, attributed to localisation of the coherent polaritonic state to molecular states undergoing excited state vibrational relaxation.

I think this is an important work, particularly given the central place of the SPI/MC photochrome in the development of organic ultrastrong coupling and polaritonic chemistry. (Controlling) interplay between coherent and localised states is also a critical and topical issue.

However, there are some significant issues that need to be addressed:

1. Figure 1c – is there a genuine, observable Rabi splitting here? The visibility of these ‘polariton’ peaks are rather limited. Could this be interpreted rather as an intermediate coupling case?

To answer this, can the density of photochromes be increased to push the system further into the SC regime?

2. Furthermore, instead of the short axis plasmon excitation serving as a weakly coupled model system, could the concentration of MC not be varied for long axis excitation? Moving smoothly from few MC to lots of MC, therefore going from weak to strong coupling, all with the long axis plasmon excitation results being the focus?

3. Both the weak and strong coupled states have reduced lifetimes compared to the bare molecular system. What is being enhanced here, non-radiative or radiative processes? And does this vary for the two cases (the strongly coupled system decays slightly faster). In other words, what is the fluorescence emission intensity/quantum yield from these systems? Is it (and hence also the radiative rate) enhanced or decreased in the various situations?

4. It was unclear to me whether collapse of the coherent state to a localised molecular state was due to molecular excited vibrational relaxation, or if the latter just followed on from the coherent state collapse?

(my intuition is that the vibrational relaxation (red shift) brings the localised excitations into resonance with the lower polariton state (present due to the N-1 molecules interacting with the plasmon), resulting in Purcell-enhanced decay of the localized excitation, but I don't know if that matches the authors thoughts)

5. "From the experiment, we find a maximum spectral shift of +0.25 eV, consistent with the value obtained theoretically (+0.4 eV) from full structural relaxation of the S1 state (see also Supplementary Fig. S6)." How does this compare to the Stokes shift (Stokes shift/2)?

6. Hutchinson -> Hutchison

Reviewer #1 (Remarks to the Author):

We thank the Reviewer for stating that “*the manuscript is in general well written, and I believe presents a very interesting and original contribution to the field*”. Below, we provide our point-by-point answers to his/her/their comments.

Reviewer’s comment. *The introduction refers only to plasmonics as a mechanism to confine the electromagnetic field beyond the diffraction limit to boost light-emitter interactions. I suggest that the authors also briefly mention relevant literature on dielectric nanoantennas in this area (see for example, Nano Lett. 16, 8, 5143–5151, 2016; Nano Lett. 17, 1219–1225, 2017).*

Authors’ answer. We agree with the Reviewer that mentioning the use of dielectric nanoantennas in the context of the topic treated in this work is relevant.

Action taken. We added a sentence in the introduction referring to the works suggested by the Reviewer (see lines 53-59), which are now new Refs [11,12] in the main text.

Reviewer’s comment. *An important experiment missing is the dynamic response of the bare plasmonic nanoantennas, as pump-probe results are shown only for the hybrid system or the bare molecular film. It is important to have separately characterized all individual elements composing the hybrid system. Can the authors comment on this?*

Authors’ answer. We agree with the Reviewer that this information is indeed important to support the claim that the observed effects really stem from the interaction between the molecular and plasmonic systems. Since the resonance of the antennas without SP falls outside the spectral region probed in our experiments, we measured the response of the antennas when SP is not yet converted into MC. Since there is no observable dynamics of the molecules in their SP form (compare Figure S18 panels b and c), this allows to retrieve the antennas’ response in their realistic environment.

Action taken. We added a sentence in the main text referring to the fact that antennas with SP do not show any significant signal (see lines 151-152 and 184-189) and added a related discussion and figure in the SI (see new Supplementary subsection 3.3 and new Figure S20)

Reviewer’s comment. *Does the signal of the hybrid system degrade over time? (On a timescale of minutes, hours, days...), i.e., because of light induced damage or permanent modification of the molecule.*

Authors’ answer. We agree with the Reviewer that this information is very much relevant. Indeed, the signal of the hybrid system degrades due to permanent modification of the molecules, apparent as a permanent color change of the sample on a time scales of minutes to hours. However, since only a small fraction of the sample is irradiated, the sample can be moved slightly to obtain measurements on the pristine molecules.

Action taken. We added a discussion on the degradation of the sample in the SI (subsection 3.1).

Reviewer's comment. *In reference to Figure 3c, the authors affirm that “theory can reproduce well both the positive-negative-positive motive of the experimental transient spectra and the slight blue shift of the negative peak at increasing Δt ”. However, I cannot see any blueshift in this figure. I suggest that the authors explicitly mention and compare in the main text the magnitude of the experimental and theoretical shifts.*

Authors' answer. We thank the Reviewer for pointing out this lack of clarity from our side. We updated Fig. 3 for a clearer presentation of the dynamics. Moreover, the numbers are now included in the main text. We obtain a blueshift of around 20 meV in the experiment and 25 meV in the simulation.

Action taken. We added a sentence in the main text where we quantify the shift (see lines 237-241).

Reviewer's comment. *Regarding Figure 3e, the authors state that “there is no significant spectral evolution of the signal over time”. However, it can be clearly seen that the spectral shape does significantly change over time. This sentence should be better expressed.*

Authors' answer. We thank the Reviewer for highlighting this point. While the Reviewer is correct that there are changes in the spectral shape over time, no pronounced spectral shift of the transient signal can be identified, in accordance with our theoretical treatment.

Action taken. We added a sentence in the main text where we comment on the change of the spectral shape in the weak coupling case (see lines 281-283).

Reviewer's comment. *The authors explain the observed frequency shift in the hybrid system as a combination of excited state absorption and stimulated emission of the relaxing molecule. I suggest that they also mention the relative contributions of these two components, required to adequately describe their experimental findings.*

Authors' answer. We thank the Reviewer for highlighting this point. We agree that it is indeed important to show the different contributions separately.

Action taken. We added a sentence in the main text (see lines 254-268 with related footnote) and a discussion with a new figure in the SI (Supplementary Fig. S13) showing the relative contributions of the components that determine the transient signal.

Reviewer's comment. *Related to my previous comment, I further recommend that the authors revise the text discussing the observed energy shifts over time, as it can be complicated to follow at parts (especially in the analysis in relation to Figures 3 and 4).*

Authors' answer. We agree with the reviewer that there was lack of clarity from our side regarding the discussion on the observed energy shifts.

Action taken. We added several sentences in the main text (between lines 245-283) clarifying the origin of these shifts and the related theoretical explanation based on the schemes of Fig. 4.

Minor comments:

i) The authors refer several times in the text to Figs. 3c,f as Figs. 3c-f (note that this can be interpreted as all panels between c and f, and not just c and f).

It has been corrected, thank you for noting that.

ii) In reference to Figure 5, in the main text, it says "...blue circles in Fig. 5a...". However, there are no blue circles in Fig. 5a (only blue diamonds or grey circles).

To avoid confusion, we now include in the figure itself explicitly the probe energy (see also answer to next comment). The main text has been modified accordingly.

iii) To facilitate understanding, it would be convenient that the authors add more information of the measurement conditions to the individual graphs. For example, so that one can understand that Figure 3 a,d corresponds to mutually orthogonal polarizations without referring to the caption. The same applies to Figure 5a, which shows plots at different probe energies, but this is not specified in the graph. This can be extended to other figures in the article.

We optimized Figures 3 and 5 to increase readability, and we thank the Reviewer for pointing out this obscurity.

Reviewer #2 (Remarks to the Author):

We are grateful to the Reviewer for his/her/their statement about our work, in particular for writing “*I think this is an important work [...] in the development of organic ultrastrong coupling and polaritonic chemistry*”, and that that studying and controlling the “*interplay between coherent and localised states is also a critical and topical issue.*” We also agree with the Reviewer about the critical points raised, and we provide below our point-by-point answers and comments.

Reviewer’s comment. *Figure 1c – is there a genuine, observable Rabi splitting here? The visibility of these ‘polariton’ peaks are rather limited. Could this be interpreted rather as an intermediate coupling case? To answer this, can the density of photochromes be increased to push the system further into the SC regime?*

Authors’ answer. We thank the Reviewer for raising this insightful question. We agree that a discussion on the existence of a polaritonic state due to a strong coupling regime is indeed necessary.

Action taken. We added a discussion on that in the SI, section 2.2, just below Eq. 10. We make use of a well-known theoretical expression for quantifying the coupling regime we are dealing with. For convenience, we quote that new paragraph below:

“On a side note, we observe that the simulated spectra in the LA case (Fig. 1e) features a Rabi splitting that well reproduce the experimental one (Fig.1c) even though the simulated peaks appear to be broader. Indeed, by fitting the bare plasmonic experimental extinction spectrum with a Lorentzian function (see Fig. S11) we obtain a linewidth Γ of ≈ 0.022 au. ($\tau \approx 1.1$ fs) that is slightly smaller than the corresponding simulated value (Fig. S10). On these grounds, to quantitatively assess which coupling regime we are dealing with in the LA case (which is the case corresponding to zero detuning between plasmon energy and molecules energies), we resort to the well-known ratio(ref. 15, SI) $\frac{2\Omega_{rabi}}{\Gamma_1+\Gamma_2}$, with Ω_{rabi} being the energy separation of the polaritonic states and Γ_1, Γ_2 being the decay rates of the uncoupled system. Considering that $\Gamma_1 = \Gamma_{LA} = 0.022$ au, $\Gamma_2 = \Gamma_{MC} = 1.5 \times 10^{-4}$ au ($\tau_2 \approx 150$ fs) and that the computed Rabi splitting value is ≈ 410 meV (≈ 0.015 au), we obtain $\frac{2\Omega_{rabi}}{\Gamma_{LA}+\Gamma_{MC}} \approx 1.4$, thus corroborating the formation of a strongly coupled system in the LA case.”

In addition to that, as shown in Fig. S9 and related discussion (just above the figure location), an increase of merocyanines concentration (that in the simulations would correspond to a smaller grid step size, green curve of Supplementary Fig. S9) would indeed lead to larger Rabi splitting, and so pushing the system even more into the strong coupling regime, given the previous argumentation in the quoted paragraph.

Reviewer’s comment. *Furthermore, instead of the short axis plasmon excitation serving as a weakly coupled model system, could the concentration of MC not be varied for long axis excitation? Moving smoothly from few MC to lots of MC, therefore going from weak to strong coupling, all with the long axis plasmon excitation results being the focus?*

Authors' answer. We thank the Reviewer for raising this important point. The Reviewer probably refers to the typical scaling of strong coupling by increasing the number of emitters. While in principle what the Reviewer suggests might be possible, to couple more emitters would require changing the concentration of the sample or the dimension of the antenna. The strength and practical importance of our platform is that it allows to switch between weak and strong coupling in experimentally exactly the same system by selectively driving the longitudinal or the transverse axes plasmons, without the need to change the sample and potentially run into reproducibility issues. As such, while we believe it would be interesting to observe the scaling, probably a different experimental setup would be required.

Nevertheless, triggered by the Reviewer's comment, we decided to explore this case numerically and estimated what happens for the long-axis excitation case if we change the number of emitters. Notably, as the number of emitters decreases, the corresponding simulated transient data (Fig. S17b, blue → green curve) seem to approach the transient response that we already observed in the short axis case, thus matching with the Reviewer's thought.

Action taken. We have added two new figures, Supplementary Figs. S16-S17 showing the results of simulations related to linear and transient signal as we vary the number of emitters coupled to the long axis mode, moving from strong to weak coupling regime.

Reviewer's comment. *Both the weak and strong coupled states have reduced lifetimes compared to the bare molecular system. What is being enhanced here, non-radiative or radiative processes? And does this vary for the two cases (the strongly coupled system decays slightly faster). In other words, what is the fluorescence emission intensity/quantum yield from these systems? Is it (and hence also the radiative rate) enhanced or decreased in the various situations?*

Authors' answer. We thank the Reviewer for raising this point. Indeed, we observe an accelerated decay of the localized excitation due to interaction with the nanoantenna in both the weak and strong coupling case. Most likely both non-radiative and radiative processes are enhanced via interaction with the plasmon, however individual contributions are experimentally indistinguishable in our case. Nevertheless, we can obtain some insights on that by our theoretical model. Indeed, inspired also by the next comment, we looked at the wavefunction of the localized state during the relaxation dynamics and we interestingly noticed that even though the wavefunction features a large coefficient close to 1 on the molecule where the excitation collapsed, it presents a residual-yet-non-null component coming from the plasmon mode. Surprisingly, this small component slightly increases during the vibrational dynamics because the localized state approaches the energy of the lower polariton state of the N-1 molecules, thus matching the Reviewer's thought of the next comment. From this observation and the theoretical data, we managed to infer that both radiative and non-radiative processes are enhanced due to the plasmons, leading to the accelerated decay that is observed. A new subsection has been added to the SI (subsection 2.3) where this analysis is described in detail.

Action taken. We added a new subsection in the SI (subsection 2.3) and a mention to that in main text (lines 319-325) discussing the experimental results of Fig. 5 (main text) based on theoretical data. For convenience, we quote that paragraph below:

“The data shown in Fig.5 (main text) illustrate the presence of an accelerated molecular decay both in the weakly (Fig. 5b) and strongly coupled case (Fig.5c) compared to isolated molecules

(Fig.5a). Interestingly, by following the wavefunction composition of the localized molecular state in the LA case during the vibrational dynamics we noticed that a residual coupling with the plasmon is still present. Notably, this coupling slightly increases when the localized molecular state approaches the energy of the lower polariton state of the remaining $N-1$ molecules. On this ground, we can reasonably relate the observed accelerated decay of the localized excitation to a plasmonic effect, leading to the sub-ps decay that is shown in Fig.5b-c. Indeed, this residual plasmonic component leads to a fast decay that can be estimated to be $\Gamma_{MC^*} \approx |C_{MC^*-p}|^2 * \Gamma_{LA}$ (where C_{MC^*-p} is the coefficient of the localized state wavefunction on the LA plasmonic state, whose squared modulus is $\approx 3.5*10^{-3}$, and $\Gamma_{LA} = 0.038$ au. ($\tau \approx 0.6$ fs) is the LA plasmon decay rate, Fig. S10) which corresponds to a lifetime of ≈ 0.2 ps.

A similar reasoning for the short axis case (SA) leads to a comparable value.

Additionally, we can (although just theoretically) decompose such enhanced decay in a radiative and a non-radiative contribution. The radiative contribution can be estimated as (IEEE J Sel Top Quantum Electron, 14,1430-1440 (2008), ref 19 SI.):

$$\Gamma_{rad}^{LA} = \frac{4}{3} \frac{\omega^3}{c^3} |\vec{\mu}_{tot}|^2 \quad (13)$$

With $\vec{\mu}_{tot}$ being the total dipole of the localized state calculated as described in section 2.2, which then presents a plasmonic contribution as mentioned above, and hence intrinsically accounts for the Purcell-enhanced radiative emission. In both the LA and SA cases we obtain that $\Gamma_{rad}^{LA/SA}$ is almost an order of magnitude smaller than the corresponding value of Γ_{MC^*} , pointing to the enhanced non-radiative decay as prevailing.”

Reviewer’s comment. “It was unclear to me whether collapse of the coherent state to a localised molecular state was due to molecular excited vibrational relaxation, or if the latter just followed on from the coherent state collapse?

(my intuition is that the vibrational relaxation (red shift) brings the localised excitations into resonance with the lower polariton state (present due to the $N-1$ molecules interacting with the plasmon), resulting in Purcell-enhanced decay of the localized excitation, but I don’t know if that matches the authors thoughts)”

Authors’ answer. We thank the Reviewer for raising two extremely interesting points which made us reflect deeply on the transient dynamics that is being probed by our experiments and simulations.

As for the first part of the question related to the “collapse of the coherent state”, our interpretation is that due to the very fast plasmon dephasing (\approx few fs), the plasmonic component of the polaritonic wavefunction quickly decays after pump absorption, well below the time resolution we have, thus leading to a coherent molecular state delocalized over many molecules. This molecular state that is delocalized over different molecules will then decohere due to the interaction with the surrounding environment, leading to a statistical ensemble of different excited single molecules with probability related to squared modulus of their respective coefficient in the initial polaritonic wavefunction. Following this collapse, each excited molecule undergoes vibrational relaxation, while the remaining ones, back in their ground state, can interact with the upcoming probe promoting a new polaritonic state where $N-1$ molecules are present. In the theoretical model that we developed, we focused on the most likely situation that can happen, that is the collective excitation collapses onto the molecule that was originally contributing most to the polaritonic state.

Concerning the second part of the Reviewer's comment, we gratefully thank him/her/them as before reading his/her/their suggestion that intriguing idea of Purcell-enhanced decay of the localized excitation due to the presence of the lower polariton formed by the N-1 molecules did not occur to us. Indeed, we checked this possibility and added a discussion on that in the SI, subsection 2.3 (see also previous comment). As detailed in the answer to the previous comment, the plasmon-enhanced decay does not require resonance with the N-1 polaritonic state, yet it becomes somewhat faster approaching such resonance.

Action taken. We added a discussion on that in the new SI subsection 2.3.

Reviewer's comment. *"From the experiment, we find a maximum spectral shift of +0.25 eV, consistent with the value obtained theoretically (+0.4 eV) from full structural relaxation of the S1 state (see also Supplementary Fig. S6)." How does this compare to the Stokes shift (Stokes shift/2)?*

Authors' answer. We thank the Reviewer for pointing out this lack of clarity from our side. Indeed, we agree that the expression "from full structural relaxation..." was misleading and not clearly expressed.

Action taken. We removed that part of the sentence from main text and clarified better (see lines 181-183), as well as we better expressed in the SI (see Supplementary Fig. S6 and related discussion) that the theoretical value of +0.4eV is indeed the Stokes-shift value.

Reviewer's comment. *Hutchinson -> Hutchison*

Authors' answer. We thank the Reviewer for spotting this mistake.

Action taken. We corrected the name of the Author (see line 326).

REVIEWERS' COMMENTS

Reviewer #1 (Remarks to the Author):

I find that the manuscript has improved significantly after revision and is now ready for publication in Nature Communications.

Reviewer #2 (Remarks to the Author):

In their resubmitted manuscript, the authors have answered at least partially my previous concerns, albeit with theory rather than experiments.

I have some caveats listed below, but I broadly support publication. Transient spectroscopy of molecular polaritonic systems is extremely complex and an active area of research of itself, and the details of the interpretations here may be debateable and revised in the future. Nevertheless, the results in Figure 5, showing an enhanced excited state decay for the strongly coupled Merocyanine compared to weakly coupled and bare film cases is extremely important. Merocyanine strong coupling has been the subject of seminal works in terms of ultrastrong coupling [Ref.8], and polaritonic chemistry [Ref. 34], and the enhanced decay observed here corroborates that observed in [Ref. 34].

In terms of caveats I would ask the authors to be sure of the following points:

-again, coming back to Fig 1C, and despite the authors added justifications, I am not sure the system is absolutely in the strong coupling regime, maybe 'intermediate-to-strong coupling' could be invoked to cover this off.

-The authors present transient transmission spectroscopy, but neglect to collect transient reflection and thus transient absorption data? Delta T and Delta R have been shown to enhance different features experimentally (see Schwartz et al. DOI: 10.1002/cphc.201200734)

-The raw data is included in Figure 5b (grey) but not 5c?

-References should be checked, [ref.32] at line 319 should be [ref.34]?

Answers to Reviewers' comments

Reviewer #1

We thank the Reviewer for supporting the publication of our work by stating that he/she/they “*find that the manuscript has improved significantly after revision and is now ready for publication in Nature Communications*”.

Reviewer #2

We thank the Reviewer for carefully reading our point-by-point answers and providing some additional insightful comments. We also thank the Reviewer for stating that he/she/they “*broadly support publication*”. We are also happy that the Reviewer recognized the importance of our work in the field by stating that “*the enhanced decay observed here corroborates that observed in [Ref. 34]*”. Here below are our point-by-point answers to the remaining concerns of the Reviewer.

Reviewer's comment. *Again, coming back to Fig 1C, and despite the authors added justifications, I am not sure the system is absolutely in the strong coupling regime, maybe 'intermediate-to-strong coupling' could be invoked to cover this off.*

Authors' answer. We thank the Reviewer for raising again this important point. We agree with the Reviewer that our estimation of being in the strong coupling regime is based on a theoretical calculation and we are open to refer to the state we are observing as “on the onset” of the strong coupling regime, as previously used in the community. We think that this compromise can make everyone agreeing that, while we might not be deeply in the optimal region of the strong coupling regime, a polaritonic state (that is, a completely different state compared to that of molecules weakly coupled to the plasmon excitation) is clearly formed as corroborated by our calculations. We hope that the Reviewer would accept this terminology.

Action taken. We specified that we are at the onset of the strong coupling regime following the discussion of experimental Figure 1c, see line 119. Besides, we changed the discussion on the coupling regimes (SI 2.2, see highlighted sentences in lines 235-237 in the Related Manuscript File named “Kuttruff_SP-MC_SI_highlighted.docx”), now reading: “...thus corroborating that we are at least undoubtedly at the onset of the strong coupling regime, and proper polaritonic states are formed in the LA case.

Reviewer's comment. *The authors present transient transmission spectroscopy, but neglect to collect transient reflection and thus transient absorption data? Delta T and Delta R have been shown to enhance different features experimentally (see Schwartz et al. DOI: 10.1002/cphc.201200734).*

Authors' answer. We thank the Reviewer for this insightful comment and agree that to infer quantitatively the changes in absolute absorption, it is necessary to measure transient transmission, as well as transient reflection. Indeed, in the work by Schwartz et al., transient reflection $\Delta R/R_0$ and transient transmission $\Delta T/T_0$ enhance different spectral features, because the steady-state response $T_0(R_0)$ is modulated by multiple interference inside the cavity.

However, in our geometry such interference plays only a minor role and hence transient reflection and transmission will exhibit almost the same spectral features (with opposite sign). In particular, temporal (time scales) and spectral dynamics that are analyzed in the manuscript are not affected by choice of either transient transmission or transient reflection. Nevertheless, we appreciate this important comment and mention now explicitly in the manuscript that reflection data is necessary to obtain the transient absorption in a quantitative manner.

Action taken. We added a small discussion at Lines 190-197 and included the reference suggested by the reviewer.

Reviewer's comment. *The raw data is included in Figure 5b (grey) but not 5c?*

Authors' answer. We agree with the reviewer that it makes sense to also show the raw data of the dynamics in the strong coupling case. To keep readability of Figure 5 unimpaired, we decided to add an additional Figure to the SI, where the full dynamics is shown for the upper and lower polaritonic states.

Action taken. We added Figure S21 to the SI

Reviewer's comment. *References should be checked, [ref.32] at line 319 should be [ref.34]?*

Authors' answer. Thanks for spotting this mistake in the references.

Action taken. We corrected the references.